# Realizing symmetry-guaranteed pairs of bound states in the continuum in metasurfaces

Chloe F. Doiron [1] ✉, Igal Brener [1] & Alexander Cerjan [1] ✉

Bound states in the continuum (BICs) have received significant attention for their ability to enhance light-matter interactions across a wide range of systems, including lasers, sensors, and frequency mixers. However, many applications require degenerate or nearly degenerate high-quality factor ($Q$) modes, such as spontaneous parametric down conversion, non-linear four-wave mixing, and intra-cavity difference frequency mixing for terahertz generation. Previously, degenerate pairs of bound states in the continuum (BICs) have been created by fine-tuning the structure to engineer the degeneracy, yielding BICs that respond unpredictably to structure imperfections and material variations. Instead, using a group theoretic approach, we present a design paradigm based on six-fold rotational symmetry ($C_6$) for creating degenerate pairs of symmetry-protected BICs, whose frequency splitting and $Q$-factors can be independently and predictably controlled, yielding a complete design phase space. Using a combination of resonator and lattice deformations in silicon metasurfaces, we experimentally demonstrate the ability to tune mode spacing from 2 nm to 110 nm while simultaneously controlling $Q$-factor.

Over the last decade, all-dielectric metasurfaces have emerged as a powerful platform for enhancing light-matter interactions. Unlike their metallic counterparts, all-dielectric metasurfaces can exhibit high-quality factor ($Q$) states that can be used to increase the performance of a variety of nonlinear phenomena, such as frequency conversion[1], actively tunable optics[2–4], and single-mode lasing[5,6]. Recently, bound states in the continuum (BICs) have emerged as the starting point for design principles to create states with arbitrarily large $Q$-factors across many photonic platforms. BICs are states that possess infinite lifetimes (i.e., infinite $Q$-factors) despite having frequencies that are degenerate with the surrounding continuum of radiative/scattering channels. Since no light can enter or leave a BIC, designing a system to operate at a BIC is generally undesirable for most applications. However, the existence of a BIC in a system's parameter space guarantees a neighborhood in design space (around the BIC) where a quasi-BIC is formed and the state's $Q$-factor can be made arbitrarily large. While there are a few different methods for creating BICs in metasurfaces[7–14], protecting

a state from radiating due to its incompatible symmetry with the radiative channels in free space has proven to be a robust experimental pathway for generating BICs. This method is easily predictable and provides a simple route to tunable $Q$-factors via judicious symmetry breaking. As such, symmetry-protected BICs have been used to realize lasers[15–20], narrowband optical filters[21–24], wavefront shapers[25–27], optical sensors[28–31], frequency combs[32], and metasurfaces for harmonic generation[33,34].

Nevertheless, despite the dramatic successes of using symmetry-protected BICs to enhance nonlinear optical phenomena, the majority of these previous realizations have relied on spectrally isolated BICs, where a single high-$Q$ state is used to enhance light-matter interactions. This fundamentally limits the classes of nonlinear phenomena that can be easily realized experimentally using these systems, precluding these structural designs from being used for applications which require two modes with similar frequencies (i.e., nearly degenerate) that both possess large $Q$-factors, such as spontaneous

[1]Center for Integrated Nanotechnologies, Sandia National Laboratories, Albuquerque, NM 87185, USA. ✉e-mail: cfdoiro@sandia.gov; awcerja@sandia.gov

parametric down conversion[35–37], nonlinear four-wave mixing[38,39], four-wave mixing optical bistability[40–42], and intracavity difference frequency mixing for terahertz generation[43–45]. Although some recent studies have attempted to overcome this shortcoming through fine-tuning an accidental degeneracy between two otherwise spectrally isolated BICs[36,46–50], yielding nearly degenerate states where the frequency splitting, center frequency, and relative polarization states in such systems are extremely sensitive to material variations and fabrication imperfections, both of which yield unpredictable changes to the properties of the two states[51–53].

Here, we theoretically and experimentally demonstrate a metasurface design paradigm that enables predictable and robust control over the frequency splitting and, independently, the $Q$-factors of two otherwise degenerate metasurface states. Our design methodology is based on a triangular lattice with six-fold rotational symmetry ($C_6$), which, as we show, is the only possible rotational symmetry that allows for a metasurface to possess symmetry-guaranteed degenerate pairs of symmetry-protected BICs (when combined with reflection or time-reversal symmetry). We prove that the two independent subgroups of $C_6$ each yield a distinct method for controlling the behavior of these pairs of metasurface states: two-fold rotational symmetry ($C_2$) controls the $Q$-factors of these pairs of states, while three-fold rotational symmetry ($C_3$) controls the frequency splitting between the two states in each pair. (In this work we adopt the notation of $C_n$ referring to an element in the cyclic group, and $C_n$ referring to the group itself[54]). Thus, by judiciously breaking either or both of these symmetries, our design paradigm facilitates independent control over the $Q$-factors and frequency splitting of these pairs of states. Experimentally, we realize our system in all-dielectric metasurfaces composed of silicon resonators on fused silica substrates, and measure the frequencies and $Q$-factors of the resonances using reflectance measurements. To demonstrate the versatility of our design paradigm, we fabricate reflection-symmetric metasurfaces with many strengths of symmetry breaking to destroy the system's $C_2$ or $C_3$ symmetries by using a combination of resonator and lattice deformations. This tuning process is conceptually illustrated in Fig. 1a. Moreover, as our methodology is based solely on symmetry and does not rely upon fine-tuning the states of a structure, it is robust against material variations and fabrication imperfections that preserve symmetry; any changes in the filling fraction, thickness, or isotropic dielectric constant of the metasurface affect the two states equally, and do not result in uncontrolled changes to the states $Q$-factors or frequency splitting. In comparison, when designing accidental degeneracies through fine-tuning, the degeneracy will generally be lifted by both symmetry-preserving and symmetry-breaking imperfections. Our design paradigm not only enables robust mode engineering in photonic systems, but also provides a pathway for improving the design of a broad range of devices across a wide range of fields, including quantum light sources, RF antennas, and phonon traps in mechanical systems.

## Results

### Group theory design principles

There are two essential ingredients that are necessary to realize the symmetry-guaranteed pairs of BICs that our metasurface design paradigm is predicated upon. First, for some in-plane wavevector, $\mathbf{k}_\parallel$, the in-plane symmetry of the crystalline metasurface must be sufficiently high so as to support pairs of states that are guaranteed to be degenerate[54–56]. This requires that the metasurface's dielectric structure possess three-fold ($C_3$), four-fold ($C_4$), or six-fold ($C_6$) rotational symmetry in addition to either reflection or time-reversal symmetry. (A crystalline metasurface that only possesses the rotational symmetries of a $C_n$ cyclic group is not guaranteed to exhibit any degeneracies. However, if a $C_3$-, $C_4$-, or $C_6$-symmetric metasurface's design is also reflection symmetric, or it is comprised of materials that preserve

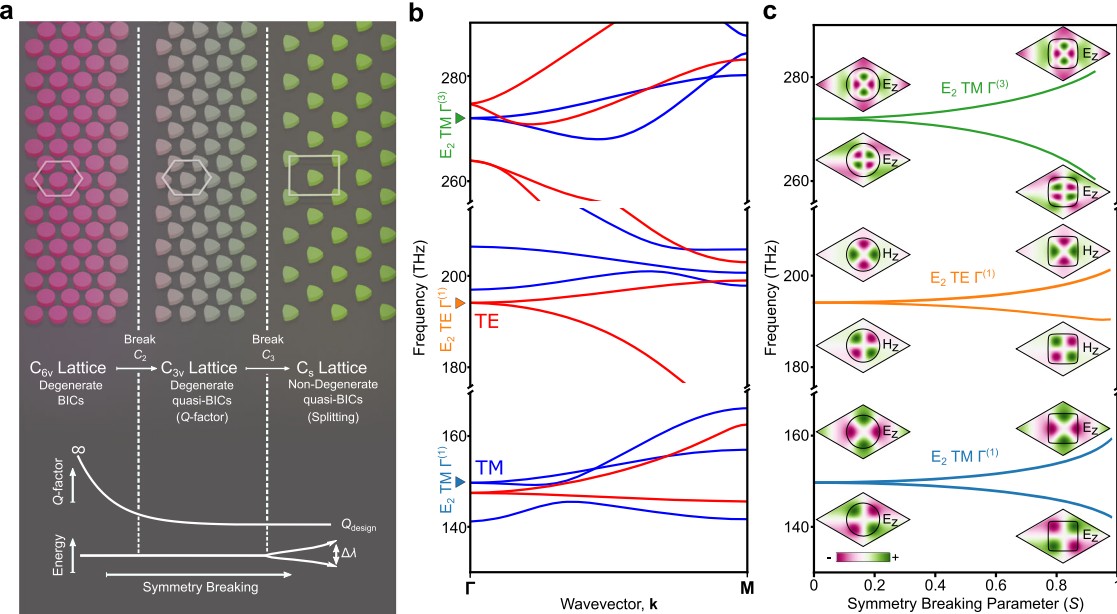

**Fig. 1 | Symmetry-protected degenerate bound states in the continuum in triangular lattices. a** Conceptual diagram of symmetry-protected degenerate bound states in the continuum. The $Q$-factor and mode splitting can be independently controlled by breaking $C_2$ and $C_3$ symmetries, respectively. **b** Band structure of two-dimensional triangular lattice of dielectric circles ($n = 3.5$). Degenerate, symmetry-protected BICs appear at the $\boldsymbol{\Gamma}$-point at 150, 195, and 270 THz denoted with triangle markers. The modes can be categorized as $E_2$ TM $\boldsymbol{\Gamma}^{(1)}$, $E_2$ TE $\boldsymbol{\Gamma}^{(1)}$, and $E_2$ TE $\boldsymbol{\Gamma}^{(3)}$, respectively. Here, the degenerate symmetry-protected BICs at $\boldsymbol{\Gamma}$ are labeled using the notation $E_2$ P $L^{(\alpha)}$. Here, P denotes whether the mode is TE or TM, L denotes the high-symmetry point in Brillouin zone, and $\alpha$ is the order of the extended Brillouin zone. Symmetry can be broken by having a non-zero in-plane wavevector, resulting in the degeneracies being lifted. **c** Eigenfrequencies for the degenerate, symmetry-protected BICs denoted in **b** when the triangular lattice of dielectric circles is transformed into a triangular lattice of dielectric squares. This transformation breaks $C_3$ lifting the degeneracy, but leaves $C_2$ intact requiring the modes to remain symmetry-protected BICs. Insets show the electric and magnetic field profiles for transverse magnetic and transverse electric modes, respectively.

**a**

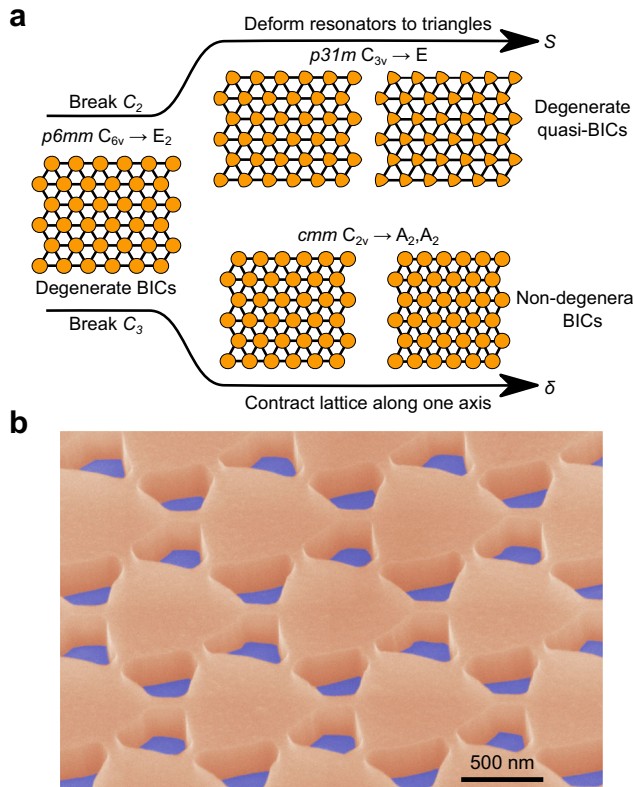

**b**

**Fig. 2 | Resonator and lattice deformations to break $C_2$ and $C_3$ symmetries. a** By deforming the circular resonators into triangles, $C_2$ symmetry can be broken (an operation that preserves $C_3$). Correspondingly, $C_3$ symmetry can be broken by contracting the lattice along one axis (an operation that preserves $C_2$). From these two deformations, we define symmetry-breaking parameters $S$ and $\delta$ to denote the strength of $C_2$ and $C_3$ symmetry breaking, respectively. **b** False-color scanning electron micrograph of a fabricated silicon metasurface on a fused silica substrate.

time-reversal symmetry, then the metasurface is guaranteed to exhibit pairs of degenerate states. When reflection symmetries are added, these degeneracies can arise from ($d \geq 2$)-dimensional irreducible representations of $C_{nv}$. When the metasurface possesses rotation and time-reversal symmetries, the Wigner criterion (Herring test) guarantees degeneracies between time-reversal conjugate representations that are otherwise one-dimensional representations.) Second, for these symmetry-guaranteed pairs of states to be BICs, they must be unable to couple to the radiative channels in the surrounding medium at the same frequency, $\omega$, and $\mathbf{k}_{\parallel}$. Although in principle there are a variety of mechanisms that can be used to prohibit a state from radiating[10], the only method that is applicable to a pair of states without fine-tuning the structure is to create a symmetry mismatch between the metasurface's states and the radiative channels in the surrounding environment. To avoid confusion between the two separate roles that symmetry is performing in our design methodology, we describe the symmetry-induced degeneracy between two metasurface states as "symmetry-guaranteed," and reserve "symmetry-protected" for the specific method of creating a BIC by prohibiting a state from coupling to the environment's radiative continuum due to its mismatched symmetry.

For a metasurface state to be a symmetry-protected BIC (regardless of whether the state is degenerate), the state must be at the center of the Brillouin zone, $\mathbf{k}_{\parallel} = \mathbf{\Gamma}$, and be even with respect to 180° rotation about the out-of-plane axis ($C_2$) or be at a momentum-space polarization singularity[57,58]. The first requirement can be proven as a consequence of the Bragg-diffraction limit and the degeneracy between s- and p-polarized waves in free space[59], while the latter

requirement stems from the fact that the radiative channels at $\mathbf{\Gamma}$ in free space (or any homogeneous and isotropic environment) are always odd with respect to $C_2$. In particular, this latter requirement is especially restrictive for symmetry-guaranteed degenerate states, as even though there are a few different possibilities of the metasurface's in-plane rotational symmetry that allow for symmetry-guaranteed degeneracies, only the presence of six-fold rotational symmetry ($C_6$) allows for degenerate pairs that are simultaneously even with respect to $C_2$ (these states form the $E_2$ irreducible representation regardless of whether reflection or time-reversal symmetry has been added to the system). Thus, only $C_6$-symmetric metasurfaces that are either reflection or time-reversal symmetric can support symmetry-guaranteed degenerate pairs of symmetry-protected BICs.

Among the possible rotational symmetry point groups of two-dimensional crystalline structures, $C_6$ is unique in possessing two independent, non-trivial subgroups, $C_2$ and $C_3$. As such, it is possible to start with a $C_6$-symmetric structure and judiciously break either its $C_2$ or $C_3$ symmetry while preserving the other (and preserving reflection and/or time-reversal symmetry). Crucially, these two possibilities for reducing the symmetry of a metasurface with $C_6$ symmetry and either reflection or time-reversal symmetry, yield independent effects on its states. For example, it is well known that breaking a metasurface's $C_2$ symmetry can be used to control the $Q$-factors of its BICs. However, breaking $C_2$ symmetry of an originally $C_6$-symmetric metasurface (that is also reflection or time-reversal symmetric) need not split the degeneracy between any of the metasurface's symmetry-guaranteed degenerate states, as the resulting $C_3$-symmetric systems still possess symmetry-guaranteed degeneracies (both the $E_1$ and $E_2$ irreducible representations of the original metasurface correspond to the E irreducible representation of the $C_3$-symmetric metasurface). Similarly, breaking $C_3$ symmetry of a $C_6$-symmetric metasurface does not change any of the states behavior under $C_2$; those states which were originally even about this rotational operation (and thus were BICs) are still even in the metasurface with reduced symmetry. Thus, the symmetry-guaranteed degenerate pairs of symmetry-protected BICs in $C_6$-symmetric metasurfaces can be controlled along two independent axes, with $C_2$ symmetry controlling the states' $Q$-factors and $C_3$ symmetry controlling the frequency splitting between these pairs of states.

Altogether, these two independent controls over the behavior of a metasurface with $C_6$ symmetry and either reflection or time-reversal symmetry are shown in Fig. 1a, which schematically shows the behavior of the $Q$-factors and frequencies of the modes as first $C_2$ symmetry is broken, and then $C_3$. A specific example in a $C_6$-symmetric metasurface with reflection symmetry ($C_{6v}$) is shown in Fig. 1b and c. Away from $\mathbf{\Gamma}$, the degeneracy between the pairs of states is broken by the non-zero wavevector (Fig. 1b). However, the degeneracy between these pairs of states can also be lifted at $\mathbf{\Gamma}$ by reducing the symmetry of the metasurface to $C_{2v}$ through geometric deformations, Fig. 1c. Here, we define a function of a single symmetry-breaking parameter, $S$, to deform the resonator cross sections from circles ($S = 0$) into squares ($S = 1$) that preserves the filling fraction of the unit cell (geometric definition in Methods). Throughout this deformation process, these modes are still BICs, as they remain even with respect to $C_2$.

## Independent control of splitting and $Q$-factor

To demonstrate full control of $Q$-factor and splitting, we designed and fabricated a series of silicon metasurfaces on fused silica substrates where two classes of deformations were applied to break the $C_2$ and $C_3$ symmetries of a lattice that would otherwise possess $C_{6v}$ symmetry. The silicon metasurfaces had a period−1000 nm, thickness−200 nm, resonator diameter−800 nm, and bar width−100 nm. While there are many combinations of resonator and lattice deformations capable of breaking $C_2$ and $C_3$ symmetries, a resonator and lattice deformation were chosen to break $C_2$ and $C_3$, respectively, since the combination is more compatible with the capabilities of electron-beam lithography.

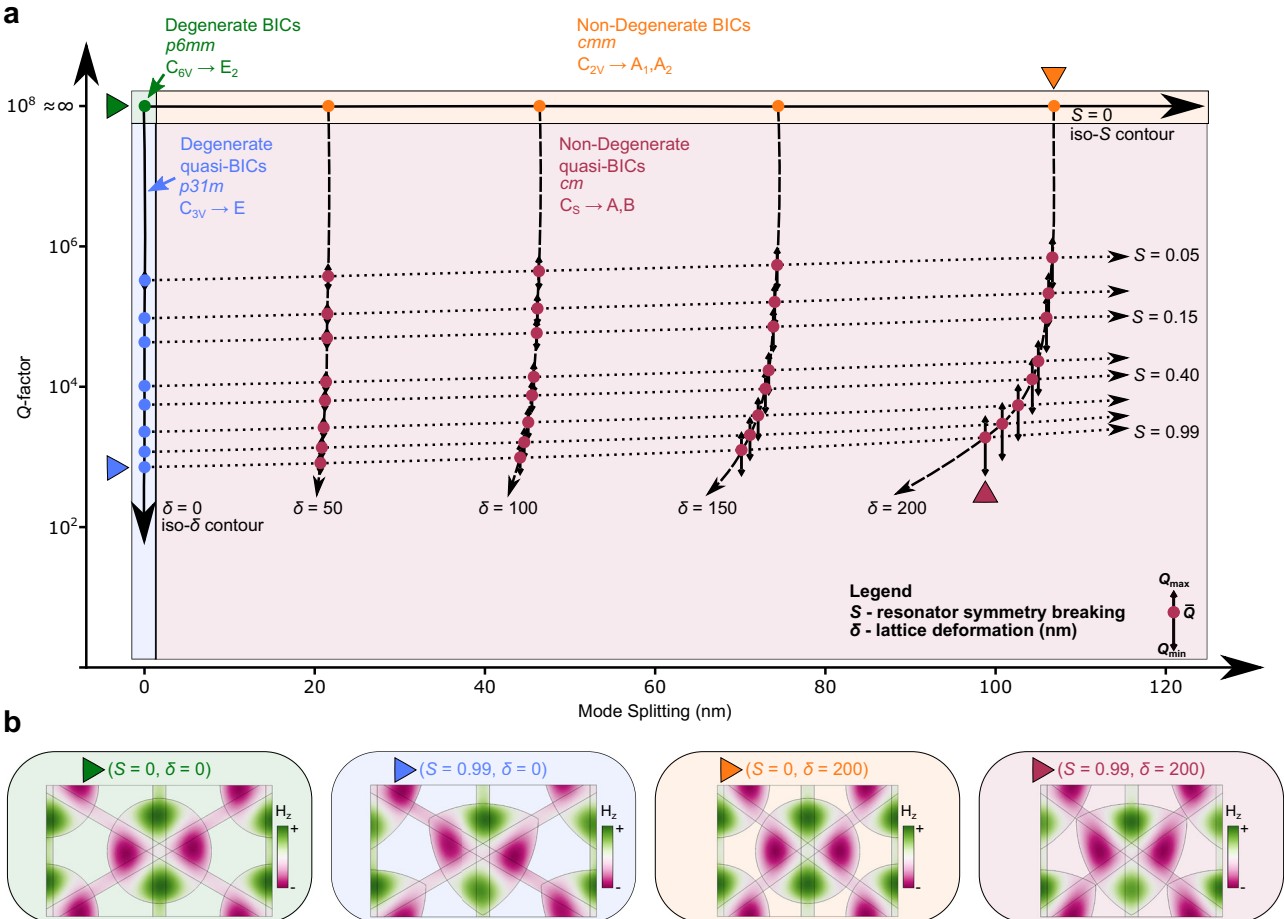

**Fig. 3 | Symmetry phases for engineering splitting and $Q$-factors. a** Symmetry phase diagram for deforming resonators into triangles and contracting the lattice. The deformations determine the splitting and $Q$-factors for the $E_2^{(1)}$ modes of a silicon metasurface. When the resonator is deformed into a triangle, only $C_2$ symmetry is broken resulting in two degenerate quasi-BICs. The $Q$-factors can be controlled by the resonator symmetry-breaking parameter ($S$). Correspondingly, when the lattice is contracted in one direction two non-degenerate BICs are formed since only $C_3$ symmetry is broken. With these two symmetry-breaking parameters it is possible to design a lattice with $cm$ $C_s$ symmetry capable of having non-degenerate quasi-BICs with arbitrary splitting and $Q$-factors. **b** Field profiles for the four symmetry phases.

An extended discussion on design considerations and an experimental demonstration using only resonator deformations to break $C_2$ and $C_3$ symmetries are presented in the supplementary information.

To break $C_2$ symmetry, the circular resonators were smoothly deformed into triangular resonators (geometric definition in Methods) as depicted in Fig. 2a (this operation preserves $C_3$ and reflection symmetries, as well as the filling fraction of the unit cell). In contrast, the lattice was contracted along one axis to break $C_3$ symmetry (an operation that preserves $C_2$ and reflection symmetries). From these deformations, we define the $C_2$ and $C_3$ symmetry-breaking parameters as $S$ and $\delta$, respectively. A false-color scanning electron micrograph of a representative metasurface is presented in Fig. 2b.

Together, these symmetry-breaking parameters enable navigation through four symmetry phases and therefore allow control over $Q$-factor and mode spacing. The symmetry phase diagram for pairs of originally degenerate, symmetry-protected BICs is presented in Fig. 3. The values for $Q$-factors and mode splitting correspond to the eigenvalue simulation results for the $E_2^{(1)}$ modes of a silicon metasurface. Here, the notation $E_2^{(n)}$ denotes the $n$th pair of symmetry-protected degenerate BICs. As discussed, when no symmetry breaking is present the modes making up the $E_2$ irreducible representation of the $p6mm$ triangular lattice form a pair of degenerate, symmetry-protected BICs, and therefore are fully decoupled from free space ($Q = \infty$).

A new symmetry phase occurs when only resonator deformations are present ($\delta = 0$ nm and $S > 0$). Since the triangular resonators preserve $C_3$ symmetry, the lattice is reduced to the $p31m$ space group which poses a symmetry-guaranteed degeneracy. Combining this symmetry guarantee with the fact that the metasurface is not $C_2$ symmetric, we can characterize this symmetry phase as having degenerate quasi-BICs. (Specifically, the modes making up the $E_2$ irreducible representation are continued into the E irreducible representation). As the deformation is increased, the $Q$-factor can be continuously decreased. Increasing the resonator deformation parameter to 0.05 causes the calculated $Q$-factor to drop from $\infty$ to $3.3 \times 10^5$. Increasing the resonator deformation parameter even further to 0.99 results in a $Q$-factor of 714.

A separate symmetry phase occurs when the lattice is contracted along one axis ($\delta > 0$ nm and $S = 0$) since the triangular lattice is reduced to a lattice belonging to the $cmm$ space group. For this space group, the modes of the $E_2$ irreducible representation are reduced into non-degenerate symmetry-protected BICs (even with respect to $C_2$). (The modes making up the $E_2$ irreducible representation are continued into the one-dimensional $A_1$ and $A_2$ irreducible representations.) As the degeneracy is no longer protected by symmetry, the frequencies of the modes begin to split. Therefore, this symmetry phase can be characterized as having two non-degenerate BICs. Eigenvalue simulations predict that by increasing the lattice deformation from 50 to

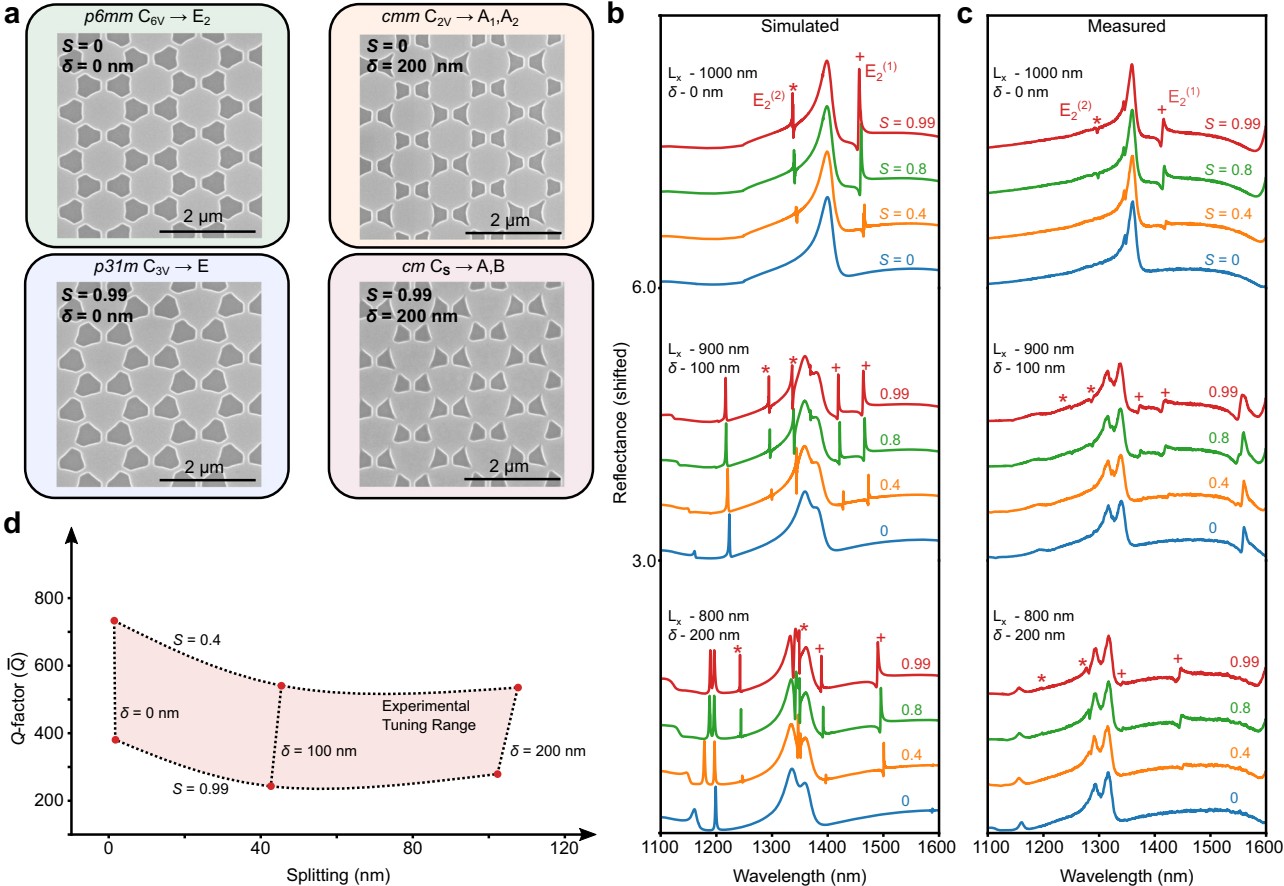

**Fig. 4 | Resonator and lattice deformations to engineer splitting and $Q$-factor.**
**a** Scanning electron micrographs of fabricated silicon metasurfaces with designs
for the four regimes of resonator deformation into a triangle and/or lattice con-
traction. Simulations (**b**) and measurements (**c**) of unpolarized reflectance for
resonator and lattice deformations of the silicon metasurface. When no lattice
deformation is present ($\delta = 0$ nm) and the resonators are cylinders ($S = 0$) no modes
are visible. If the resonators are deformed into triangles two peaks appear from

degenerate quasi-BICs from the $E_2^{(1)}$ and $E_2^{(2)}$ modes. If the lattice is deformed
($\delta > 0$ nm) the degeneracy of the quasi-BICs is lifted allowing four modes to be
observed. As the deformation is increased the mode splitting increases. At any
deformation, the resonators can be deformed into cylinders ($S = 0$) which forms
two sets of non-degenerate BICs. Since these modes are true BICs they are fully
decoupled from the far-field and no longer observable. **d** Experimental mode
splitting and $Q$-factor tuning range from reflectance measurements.

200 nm the mode splitting can be increased from 22 to 107 nm
demonstrating the continuous tunability of mode splitting in this
symmetry phase using lattice contraction.

When both lattice and resonator deformations are present
($\delta > 0$ nm and $S > 0$) the triangular lattice is reduced to a lattice
belonging to the *cm* space group. Under this symmetry reduction, the
degeneracy is no longer guaranteed by symmetry and the modes can
couple to the radiative continuum in the environment. This means that
the modes in this phase can be characterized as non-degenerate quasi-
BICs with $Q$-factors and splitting controlled by lattice and resonator
deformations respectively. (Specifically, the $E_2$ irreducible repre-
sentation is reduced into the one-dimensional A and B irreducible
representations). Since both $C_2$ and $C_3$ rotational symmetries are
broken, the two quasi-BICs are no longer required to have the same $Q$-
factors. Because of this, Fig. 3a presents the mean, maximum, and
minimum $Q$-factors for the two $E_2^{(1)}$ modes of a silicon metasurfaces. As
the lattice deformation is increased the difference between the $Q$-
factors increases.

To corroborate the numerical predictions from eigenvalue simu-
lations, we performed near-normal incidence reflectance measure-
ments using unpolarized light on the series of metasurfaces with
symmetry-breaking parameters spanning the four symmetry phases.
Representative scanning electron micrographs for the four symmetry
phases are presented in Fig. 4a. As predicted, when no symmetry

breaking was present ($\delta = 0$ nm and $S = 0$) the $E_2^{(1)}$ and $E_2^{(2)}$ modes were
not observable in simulated and measured reflectance measurements
(Fig. 4b, c). Furthermore, when no lattice deformation was present and
the resonators were deformed ($S > 0$) the $E^{(1)}$ and $E^{(2)}$ modes became
observable in simulated and measured reflectance spectra (Fig. 4b, c).
By increasing the strength of the resonator deformation parameter ($S$)
the peak reflectance of the degenerate quasi-BICs increased, but at the
cost of reduced $Q$-factor. In this experimental demonstration, we
observed a maximum $Q$-factor of 730 at $\delta = 0$ nm and $S = 0.4$. The
observed $Q$-factor decreased to 380 when the resonator deformation
parameter was increased to 0.99. Compared to simulations, the
observed $Q$-factors were below predicted values. Furthermore, the
experimental $Q$-factor tuning range was significantly smaller than the
predicted tuning range. Both of these discrepancies can be attributed
to radiative damping from imperfection induced symmetry breaking
and non-radiative damping. The measured $Q$-factors are consistent
with an imperfection and non-radiative limited $Q$-factor $\approx 840$. Never-
theless, our experimental measurements demonstrate that $C_2$ sym-
metry breaking and $C_3$ symmetry-preserving resonator deformations
allow the $Q$-factor to be engineered while maintaining the symmetry-
guaranteed modal degeneracy. When only lattice deformations were
present ($\delta > 0$ nm and $S = 0$) no splitting could be measured from
simulated or measured reflectance spectra (Fig. 4b, c) since the modes
are non-degenerate BICs that are fully decoupled from free space.

Consequently, both resonator and lattice deformations were necessary to experimentally confirm predicted mode splitting.

Using the experimental reflectance measurements, we retrieved the $Q$-factors and mode splittings to characterize the experimental tunability range. The experimental tunability range is shown in Fig. 4d. By controlling the resonator and lattice deformations we tuned the $Q$-factor by almost a factor of two and the mode splitting from 2 to 110 nm. Compared to simulations, the average mode wavelength differed by 47 nm, but the mode splitting only differed from simulations by 2 nm on average, with a maximum deviation of 4 nm.

Combined together, the mode splitting and $Q$-factor tunability illustrate the potential for symmetry-protected degenerate BICs to enable the design of arbitrarily close high-$Q$ modes. In particular, this experimental demonstration validates one of the main features of our design paradigm, that the rotational symmetry of the system guarantees the degeneracy of the metasurface states without requiring any fine-tuning. Even though there are discrepancies between the central frequency of the simulated and experimentally observed systems, the frequency splitting is predictable and robust. This robustness stands in stark contrast to the fragility of accidental degeneracies where there is no general guarantee for how the two accidentally degenerate BICs respond to perturbations. For accidental degeneracies there are two optimal responses to perturbations that can occur, one where the frequencies of both modes shift together equally (preserving the degeneracy) and another where the frequencies shift equally in opposite directions (preserving the center frequency). These optimal responses are not guaranteed and can only result from fine-tuning. Therefore, in general neither the degeneracy nor center frequency will be preserved when accidentally degenerate BICs are perturbed. The robustness of symmetry-protected degenerate BICs arises from the fact that both modes are symmetry-guaranteed to shift equally when the system is perturbed in a way that preserves $C_3$ symmetry (i.e., isotropic refractive index, resonator thickness, or resonator diameter). This robustness combined with independent control of $Q$-factor and splitting, make symmetry-protected degenerate BICs an ideal starting point for designing metasurfaces with arbitrarily close high-$Q$ modes.

## Discussion

In summary, we have developed a general approach using principles rooted in group theory to create BICs where not only are lifetimes (and thus $Q$-factors) protected by symmetry but also degeneracies leading to the creation of symmetry-protected degenerate BICs. Based on this design, we experimentally proved that symmetry-protected degenerate BICs facilitate the design of high-$Q$ modes with arbitrary frequency splitting. Furthermore, we experimentally demonstrated the ability to control $Q$-factor and frequency splitting through two classes of symmetry deformations. Moreover, our experimental demonstration provides explicit verification of one of the main advantages of our approach—fabrication imperfections only yielded significant changes to the average properties of both modes, not their relative properties. While we implemented this design paradigm in dielectric metasurfaces, it can be extended to other nanophotonic material platforms. For example, applying our design methodology to metasurfaces with high second-order susceptibilities has the potential to enable the creation of complex quantum states through SPDC[37]. Further extensions to our design paradigm may enable enhanced polarization control (specifically chirality) by breaking the remaining mirror symmetry planes. This chirality may lead to new methods for designing single-fed circularly polarized RF antennas[60]. These extensions have the potential to increase the dimensionality of symmetry phase diagrams allowing for increased control over the optical responses of metasurfaces. Looking beyond optical metasurfaces, our design paradigm provides a path for the creation of near degenerate mechanical BICs for phonon trapping[61–63]. Altogether, these future directions show the potential for

using symmetry-based approaches for creating and controlling structures with nearly degenerate BICs.

## Methods

### Near-normal reflectance measurements

We used a homemade near-IR microscope for near-normal reflectance measurements. The illumination source was a stabilized tungsten halogen lamp. To illuminate and collect at low angles of incidence we used a 1.25 mm aperture to reduce the entrance pupil of a 50 mm focal length plano-convex lens leading to a NA of 0.015 with an angle of incidence less than 1°. The collected reflectance spectrum was directed via a fiber to the entrance slit of a grating spectrometer (Acton, Spectra Pro 2500i). We used a liquid nitrogen cooled InGaAs photodiode array to measure reflectance spectra. We used the reflectance spectrum from a silver mirror to normalize all measured reflectance spectra.

### Nanofabrication

The fabrication of silicon metasurfaces began with JGS2 grade fused silica substrates. We deposited a thin film of $Al_2O_3$ (10 nm) and silicon (200 nm) through electron-beam evaporation. The $Al_2O_3$ layer was used an etch stop layer. The samples were then annealed using a rapid thermal anneal (Jipelec, Jetfirst) at 900 °C for 5 min to reduce optical losses in the thin film. An electron-beam resist (ZEP520A) was deposited by spin coating for electron-beam lithography. An anti-charging coating (DisChem Inc., DisCharge) was applied by spin coating to prevent charging during lithography. A 100 keV electron-beam lithography system (JEOL, JBX-6300FS) was used for pattern generation with the ZEP520A pattern developed by n-amyl acetate for 2 min. The pattern was transferred to the silicon through reactive ion etching (PlasmaTherm, SLR) using a mixture of $SF_6$ (33 sccm) and $C_4F_8$ (77 sccm) at a working pressure of 10 mT with capacitive and inductive powers of 20 and 750 W, respectively. The remaining resist was removed with hot (60 °C) NMP for 1 h followed by a final $O_2$ plasma descum.

### Electromagnetic simulations

All full-wave electromagnetic eigenvalue simulations were performed with the finite element method using COMSOL Multiphysics. Far-field reflectance simulations were performed using finite-difference time-domain (FDTD) simulations using Lumerical FDTD.

### Symmetry-breaking deformations

The deformation from a circle to a square is given by Equation (1). To minimize the shifting of the modes while tuning the symmetry-breaking parameter, the diameters was scaled using the area of the full-circle to full-square deformation with the same symmetry-breaking parameter.

$$y(x,S) = \pm \frac{\sqrt{x^2 - r^2}}{\sqrt{\frac{S^2 \cdot x^2}{r^2} - 1}}, \text{for } -r \leq x \leq r \qquad (1)$$

The deformation from a circle to a triangle is given by Equation (2). To minimize the shifting of the modes while tuning the symmetry-breaking parameter, the diameter was scaled to keep the area constant as a function of symmetry-breaking parameter.

$$r(\theta, S) = r \cdot \left[ 1 - S + S \cdot \left( \cos\left(\theta - \frac{2\pi \text{floor}\left(\frac{3 \cdot (\theta - \pi)}{2\pi} + 1/2\right)}{3}\right) \right. \right.$$
$$\left. \left. + \left[ 1 + 2 \cdot \cos\left(\frac{\pi}{3}\right) + \cos\left(\theta - \frac{2\pi \text{floor}\left(\frac{3 \cdot (\theta - \pi)}{2\pi} + 1/2\right)}{3}\right) \right]^2 \right)^{1/2} \right]$$
$$(2)$$

## Data availability

The data that support the plots within this paper and other findings of this study are available from the corresponding authors upon reasonable request.

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

## Acknowledgements
I.B. and C.F.D. acknowledge support from the U.S. Department of Energy, Office of Basic Energy Sciences, Division of Materials Sciences and Engineering (BES 20-017574 I.B.). A.C. acknowledges support from the Laboratory Directed Research and Development program at Sandia National Laboratories. This work was performed, in part, at the Center for Integrated Nanotechnologies, an Office of Science User Facility operated for the U.S. Department of Energy (DOE) Office of Science. Sandia National Laboratories is a multi-mission laboratory managed and operated by National Technology and Engineering Solutions of Sandia, LLC, a wholly owned subsidiary of Honeywell International, Inc., for the U.S. Department of Energy's National Nuclear Security Administration under contract DE-NA0003525. This paper describes objective technical results and analysis. Any subjective views or opinions that might be expressed in the paper do not necessarily represent the views of the U.S. Department of Energy or the United States Government.

## Author contributions
A.C. and C.F.D. initiated this research. C.F.D. designed, fabricated, and characterized the samples. All authors analyzed and interpreted data. A.C. and I.B. supervised the project. C.F.D. and A.C. wrote the manuscript with input from all authors.

## Competing interests
The authors declare no competing interests.
