## [Peer review file · Nature Communications]

REVIEWER COMMENTS

Reviewer #1 (Remarks to the Author):

I find the paper by C.F. Doiron, I. Brener, and A. Cerjan extremely interesting and pioneering. Additional solid argument in favour of this paper is experimental verification of the theory.

Starting point is based on metasurface with the symmetry C_6 which is constituted of two symmetry subgroups C_2 and C_3 .

That allows existence of symmetry-guaranteed degenerate pairs of symmetry-protected BICs which belong to these subgroup symmetry groups. That is the mainstream idea of the paper.

Comments

1. Defect localized BICs in photonic crystals were predicted long time before the paper [12] by Bulgakov and Sadreev, PRB 78, 075105 (2008). That paper stimulated experiments by Segev's group cited as [8] for revealing symmetry protected BICs in photonic crystal.

2. The authors state in page 3 (lines 69-72) that "as our methodology is based solely on symmetry and does not rely upon fine-tuning the states of a structure, it is robust against material variations and fabrication imperfections; any changes

in the filling fraction (fabrication imperfection) or dielectric constant (material variation) of the metasurface affects the two states equally, and does not result in uncontrolled changes to the states Q-factors or frequency splitting." However, first of all material fluctuations break the symmetry C_{3v} and C_3 that removes degeneracy of BICs classified according to irreducible representations of group symmetry. It would be nice if the authors argue their statements more clearly. Anyway I believe the structural imperfections are to be small enough in order to guarantee pairs of degenerate quasi-BICs.

3. Why the simple rotational groups C_2 and C_3 are considered by the authors as nontrivial groups. Any group C_n is the Abelian group with one-dimensional irreducible representations. Maybe the authors imply the group C_{3v} which has one two-dimensional irreducible representation E and therefore modes can be two-fold degenerate. Respectively, already they can be symmetry-guaranteed pairs of the BICs.

4. Could the authors present more solid arguments in favor of that the degenerated BICs widely limit the classes of non-linear phenomena?

Reviewer #2 (Remarks to the Author):

The authors of manuscript “Realizing symmetry-guaranteed pairs of bound states in the continuum in metasurfaces” suggest a design of metasurface supporting bound states in the continuum (BIC) protected by symmetry. Based on the group theoretical analysis they consider a high-symmetry structure supporting BIC. Reducing of different type of symmetry allows one to control either quality (Q) factor or mode splitting. I like the underlying idea very much and in general it suitable to be a publication in NCOMM. In the present form it is really far of it, though.

1. Authors do not reveal they distinguish point symmetry of scatterer and lattice symmetry. For example, the transformation of the disk shape into triangle they consider as C2 breaking. At the same time the lattice deformation they treat as C3 breaking. Although they really are, these two modifications of the metasurface have very different nature. I easily imagine change of the scatterer shape that reduces any of the considered symmetry and the same reduction related to the lattice deformation as well. I expected that the former leads to the control of Q factor, and the latter is responsible to the mode splitting. I believe it is the most important results of the manuscript and these two effects authors have to elaborate in great details and illustrations which mechanism results in which effect.

2. Starting from Section “Independent control of splitting and Q-factor” theoretical and experimental results are described in confusion way. A small part of text is to experiment and the very next part is theory without any warning and right at once it goes back to the experiment and so on. It misleads a reader and has to be rewritten in a more structuralized way.

3. Oblique incidence is known to cause reduction of the symmetry. Is a one degree convergence is negligible. The authors have to carry out proper simulations sustaining this assumption.

4. The manuscript contains vast of the group-symmetry terms, which are in excessive way and sometimes misleading. I recommend the authors to give their manuscript to an expert in the group theory analysis applied to condensed matters to improve the terminology.

5. Several times the authors claim on benefits over the problem of fabrication imperfections, however they refer to no studies. For example, recent paper in Nanophotonics 10, 4313 (2021) examines the robustness of the BIC of different type against the typical fabrication imperfectness, that is positional disorder. I recommend the authors to consider that paper and references therein.

Overall, I like the idea but the present form of the manuscript makes me unhappy. My recommendation is a major revision mainly to improve the presentation of the results and to make it clearer.

Reviewer #3 (Remarks to the Author):

In the manuscript, Authors discuss the origin of bound states in the continuum, high-Q resonances of periodic photonic structures, in terms of group theory. In particular, they present an approach to create two degenerate symmetry-protected bound states in the continuum and control their quality factor. It is a beautiful example of rigorous theoretical analysis which results can be implemented in practical applications.

Please, find comments and questions below.

Line 145 seems to have a misprint.

In Fig.3(a) diagram shows different regimes of symmetry breaking. How was the quality factor obtained? Highly-likely, from simulation, but why did the Authors take average values of quality factor?

Could the Authors comment on how the orientation of the axis along which the lattice is squeezed, is chosen? Is it restricted by basis vectors of the primitive unit cell? Will there be any quantitative differences in Qs and frequencies splitting compared to the discussed case?

Authors' Response to Reviews of

Realizing symmetry-guaranteed pairs of bound states in the continuum in metasurfaces

Chloe F. Doiron, Igal Brener, and Alexander Cerjan
Nature Communications,

RC: Reviewers' Comment, AR: Authors' Response, □ Manuscript Text

1. Reviewer #1

RC: *I find the paper by C.F. Doiron, I. Brener, and A. Cerjan extremely interesting and pioneering. Additional solid argument in favour of this paper is experimental verification of the theory. Starting point is based on metasurface with the symmetry C_6 which is constituted of two symmetry subgroups C_2 and C_3 . That allows existence of symmetry-guaranteed degenerate pairs of symmetry-protected BICs which belong to these subgroup symmetry groups. That is the mainstream idea of the paper.*

AR: We want to begin by thanking the reviewer for their time and constructive feedback that enabled us to significantly improve the manuscript.

1.1. Comment #1

RC: *Defect localized BICs in photonic crystals were predicted long time before the paper [12] by Bulgakov and Sadreev, PRB 78, 075105 (2008). That paper stimulated experiments by Segev's group cited as [8] for revealing symmetry protected BICs in photonic crystal.*

AR: Thank you for pointing out this omission. We have added the work by Bulgakov and Sadreev to the references as [13] and updated the citation order to better reflect the timeline for the development of BICs.

Since no light can enter or leave a BIC, designing a system to operate at a BIC is generally undesirable for most applications. However, the existence of a BIC in a system's parameter space guarantees a neighborhood in design space (around the BIC) where a quasi-BIC is formed and the state's Q -factor can be made arbitrarily large. While there are a few different methods for creating BICs in metasurfaces^{7-12,7-14}, protecting a state from radiating due to its incompatible symmetry with the radiative channels in free space has proven to be a robust experimental pathway for ~~creating~~ generating BICs.

1.2. Comment #2

RC: *The authors state in page 3 (lines 69-72) that "as our methodology is based solely on symmetry and does not rely upon fine-tuning the states of a structure, it is robust against material variations and fabrication imperfections; any changes in the filling fraction (fabrication imperfection) or dielectric constant (material variation) of the metasurface affects the two states equally, and does not result in uncontrolled changes to the states Q -factors or frequency splitting." However, first of all material fluctuations break the symmetry C_{3v} and C_3 that removes degeneracy of BICs classified according to irreducible presentations of group symmetry. It would nice if the authors argue their statements more clearly. Anyway I believe the structural*

imperfections are to be small enough in order to guarantee pairs of degenerate quasi-BICs.

AR: Thank you for identifying this compact wording resulting in a loss of clarity. Both fabrication and material variations can result in symmetry preserving imperfections or symmetry breaking imperfections. We intended to highlight the fact that there are many types of common material and fabrication imperfections that do not break C_{3v} or C_3 symmetries, including resonator thickness, resonator diameter, lattice periodicity, and isotropic refractive index. In comparison, when designing accidental degeneracies through fine-tuning the degeneracy will be susceptible to these imperfections in general. We have expanded the discussion in page 3 (lines 69-72) to more clearly articulate the comparison between symmetry-guaranteed and accidental degeneracies.

Moreover, as our methodology is based solely on symmetry and does not rely upon fine-tuning the states of a structure, it is robust against material variations and fabrication imperfections that preserve symmetry; any changes in the filling fraction (~~fabrication imperfection~~) or dielectric constant (~~material variation~~), thickness, or isotropic dielectric constant of the metasurface ~~affects~~ affect the two states equally, and ~~does do~~ not result in uncontrolled changes to the states Q -factors or frequency splitting. In comparison, when designing accidental degeneracies through fine-tuning, the degeneracy will generally be lifted by both symmetry-preserving and symmetry-breaking imperfections. Our design paradigm not only enables robust mode engineering in photonic systems, but also provides a pathway for improving the design of a broad range of devices across a wide range of fields, including quantum light sources, RF antennas, and phonon traps in mechanical systems.

1.3. Comment #3

RC: *Why the simple rotational groups C_2 and C_3 are considered by the authors as nontrivial groups. Any group C_n is the Abel group with one-dimensional irreducible representations. May be the authors imply the group C_{3v} which has one two-dimensional irreducible representation E and therefore modes can be two-fold degenerate. Respectively, already they can be symmetry-guaranteed pairs of the BICs.*

AR: We appreciate the reviewer pointing out our lack of clarity on this issue. Yes, any C_n group, by itself, will only possess one-dimensional irreducible representations. However, our structures use materials that are time-reversal symmetric (indeed, most materials used in nanophotonic systems have this property). Thus, the the full set symmetries of our system are described by the nonunitary group $\bar{G} = G + \mathcal{T}G$, that combines both the unitary space group operations with the antiunitary operations formed by the composition of a space group operation with \mathcal{T} , the time-reversal operator. The inclusion of time-reversal symmetry can guarantee additional degeneracies in a system's spectrum that are not present when only considering the unitary space group operators, see Chapter 13 of Ref. [1].

The net effect of time-reversal symmetry for our photonic (bosonic) crystal slab systems is to force the same degeneracies that are found in slab systems with reflection symmetries. In other words, both a photonic crystal slab with C_{3v} symmetry, and a slab with \mathcal{T} and C_3 symmetries, will possess a symmetry guaranteed degeneracy between time-reversal conjugate representations that are otherwise one-dimensional representations.

To clarify these concepts, we have substantially amended the discussion of symmetries in our manuscript.

There are two essential ingredients that are necessary to realize the symmetry-guaranteed pairs of BICs that our metasurface design paradigm is predicated upon. First, for some in-plane wavevector, k_{\parallel} , the in-plane symmetry of the crystalline metasurface must be sufficiently high so as to sup-

port pairs of states that are guaranteed to be degenerate (i.e., the little group of \mathbf{k}_{\parallel} must possess an irreducible representation with dimension 2)^{54–56}. This requires that the metasurface’s dielectric structure possess three-fold (C_3), four-fold (C_4), or six-fold (C_6) rotational symmetry in addition to either reflection or time-reversal symmetry.* (A crystalline metasurface that *only* possesses the rotational symmetries of a C_n cyclic group is not guaranteed to exhibit any degeneracies. However, if a C_3 -, C_4 -, or C_6 -symmetric metasurface’s design is also reflection symmetric, or it is comprised of materials that preserve time-reversal symmetry, then the metasurface is guaranteed to exhibit pairs of degenerate states. When reflection symmetries are added, these degeneracies can arise from ($d \geq 2$)-dimensional irreducible representations of C_{nv} . When the metasurface possesses rotation and time-reversal symmetries, the Wigner criterion (Herring test) guarantees degeneracies between time-reversal conjugate representations that are otherwise one-dimensional representations.) Second, for these symmetry-guaranteed pairs of states to be BICs, they must be unable to couple to the radiative channels in the surrounding medium at the same frequency, ω , and \mathbf{k}_{\parallel} . Although in principle there are a variety of mechanisms that can be used to prohibit a state from radiating⁷, the only method that is applicable to a pair of states without fine-tuning the structure is to create a symmetry mismatch between the metasurface’s states and the radiative channels in the surrounding environment.

1.4. Comment #4

RC: *Could the authors present more solid arguments in favor of that the degenerated BICs wide limits the classes of non-linear phenomena?*

AR: We thank the reviewer for identifying this weak argument. When designing nearly degenerate high- Q states there are four properties to control: center frequency, mode spacing, Q -factors, and polarization states. When using accidentally degenerate symmetry-protected BICs only the Q -factors are guaranteed to be protected by symmetry. This creates a design space that is very susceptible to symmetry preserving and symmetry breaking imperfections resulting in an experimental platform that while workable in theory, can become extremely challenging in practice. With symmetry-guaranteed pairs of symmetry-protected BICs, not only are Q -factors protected, but also mode spacing and relative polarization states (the modes are guaranteed to have orthogonal polarization states) leaving only the center frequency unprotected. These additional symmetry protections greatly simplify the design space and feasibility of realizing nearly degenerate high- Q states, especially for applications such as spontaneous parametric down-conversion (SPDC) where ideally all four properties would be protected [2]. We have expanded the discussion in the introduction to more clearly articulate this argument. The modified section is reproduced below for the reviewer’s convenience.

Nevertheless, despite the dramatic successes of using symmetry-protected BICs to enhance non-linear optical phenomena, the majority of these previous realizations have relied on spectrally isolated BICs, where a single high- Q state is used to enhance light-matter interactions. This fundamentally limits the classes of non-linear phenomena that can be ~~realized~~ easily realized experimentally using these systems, precluding these structural designs from being used for applications which require two modes with similar frequencies (i.e., nearly degenerate) that both possess large Q -factors, such as spontaneous parametric down conversion^{35,36}^{35–37}, non-linear four-wave mixing^{38,39}, four-wave mixing optical bistability^{40–42}, and intra-cavity difference frequency generation for terahertz generation^{43–45}. Although some recent studies have attempted to overcome this shortcoming through fine-tuning an accidental degeneracy between two otherwise spectrally isolated BICs^{36,46–50}, ~~this approach results in systems that yielding nearly degenerate states where the frequency splitting, center frequency, and relative polarization states in such systems~~ are extremely sensitive to material variations and fabrication

imperfections, both of which yield unpredictable changes in the frequency splitting between, and the Q-factors of, to the properties of the two states^{51–53}.

2. Reviewer #2

RC: *The authors of manuscript “Realizing symmetry-guaranteed pairs of bound states in the continuum in metasurfaces” suggest a design of metasurface supporting bound states in the continuum (BIC) protected by symmetry. Based on the group theoretical analysis they consider a high-symmetry structure supporting BIC. Reducing of different type of symmetry allows one to control either quality (Q) factor or mode splitting. I like the underlying idea very much and in general it suitable to be a publication in NCOMM. In the present form it is really far of it, though.*

AR: We want to thank the reviewer for their time and constructive feedback. The feedback enabled us to significantly strengthen the manuscript.

2.1. Comment #1

RC: *Authors do not reveal they distinguish point symmetry of scatterer and lattice symmetry. For example, the transformation of the disk shape into triangle they consider as C_2 breaking. At the same time the lattice deformation they treat as C_3 breaking. Although they really are, these two modifications of the metasurface have very different nature. I easily imagine change of the scatterer shape that reduces any of the considered symmetry and the same reduction related to the lattice deformation as well. I expected that the former leads to the control of Q factor, and the latter is responsible to the mode splitting. I believe it is the most important results of the manuscript and these two effects authors have to elaborate in great details and illustrations which mechanism results in which effect.*

AR: We thank the reviewer for identifying the lack of clarity in the manuscript to describe the physical mechanism used in the manuscript. For this work we consider the symmetry of the scatterer and lattice together, not separately. If we understand the reviewer correctly, they are making two points in this comment. First, that it is possible to use either the scatterer’s shape or the lattice deformation to completely reduce the system’s symmetry, or to otherwise control the system’s symmetry. We agree with this comment, we simply chose to use the scatterer’s shape and lattice deformation together for simplicity of experimental realization.

Second, the reviewer is hypothesizing that, rather than C_2 -breaking and C_3 -breaking controlling the Q-factors and mode splitting, respectively, that the Q-factor is instead completely controlled by the scatterer’s geometry (regardless of what symmetries it possesses), and the mode splitting is completely controlled by the lattice deformation. On this point, we disagree.

To clearly demonstrate that it is the two different types of symmetry-breaking that provide independent control over the modal Q-factors and frequency splitting, we have updated the manuscript and supplementary information to include simulated and/or experimental results for a total of eight cases demonstrating C_2 -breaking and C_3 -breaking controlling the Q-factors and mode splitting using scatterer and/or lattice deformations. In particular, these simulations confirm that C_2 -breaking always controls the modes’ Q-factors, regardless of whether C_2 is broken by the scatterer’s shape or a deformation of the lattice, and C_3 -breaking always controls the frequency splitting between the two resonances, again, regardless of whether changes in the scatterer or lattice is responsible for this change in the system’s symmetry.

To improve the clarity of the paper we have added a discussion to the manuscript to more clearly state the role

of C_2 and C_3 symmetry breaking operations. Furthermore, we have modified the text to more clearly state why both resonator and lattice deformations were used to control Q -factor and mode splitting respectively. Finally, we have modified the manuscript to more clearly reference the extended group theory discussion, experimental demonstration using only resonator deformations, and additional numerical simulations in the supplementary information. The added sections, modified sections, and relating figures have been reproduced here for the reviewer's convenience.

Resonator deformations to control mode spacing

Instead of using lattice deformations to break C_3 and control the mode spacing, it is possible to use resonator deformations. To illustrate this we performed full-wave eigenvalue simulations when deforming cylindrical resonators to square prisms as depicted in Fig. S7a. Through this deformation, C_6 and C_3 symmetries are broken, which yields a lattice that belongs to the $cm\bar{m}$ space group, and results in the symmetry-protected degenerate BICs becoming non-degenerate BICs. Correspondingly, as the symmetry breaking parameter is increased the mode splitting increases as shown in Fig. S7b, but since both modes are BICs the Q -factors remain $\approx \infty$ for all symmetry breaking parameters as confirmed in Fig. S7c.

Figure R1: **Circle to square deformation to control splitting** (a) Schematic illustrating deforming the cylindrical resonators in a triangular lattice to square prism resonators. This breaks C_6 and C_3 symmetries reducing the the symmetry from a lattice belonging to the $p6mm$ space group to one belonging to the $cm\bar{m}$ space group causing the symmetry-protected degenerate BICs to become non-degenerate BICs. This can be observed from the mode wavelengths (b) and Q -factors (c) as a function of symmetry breaking. As the the cylindrical resonators are deformed into square prisms the degeneracy is lifted. But since the modes are non-degenerate BICs, the modes remain BICs with Q -factors $\approx \infty$ for all symmetry breaking parameters.

Experimental observation of symmetry-protected degenerate BICs using single resonator symmetry breaking parameter to control splitting and Q -factor

Figure R2: **Experimental demonstration of symmetry-protected degenerate BICs.** (a) Wavelength and Q -factor for the $E_2^{(1)}$ modes from full-wave simulations for half-square, half-circle silicon resonators in a triangular lattice on a fused silica substrate. When the symmetry breaking parameter is zero (cylindrical resonators) the $E_2^{(1)}$ modes are degenerate BICs. As the symmetry breaking is increased the modes split and the Q -factor decreases as the modes become non-degenerate quasi-BICs. (b) Field profiles of H_z for $E_2^{(1)}$ modes with cylindrical resonators ($S=0$) and half-square, half-circle resonators ($S=0.99$). (c) Scanning electron micrographs of fabricated metasurfaces with symmetry breaking parameters from $S=0$ to $S=0.99$. Full-wave simulations (d) and experimental measurements (e) of reflectance of a silicon metasurface. As the symmetry breaking parameter is increased from zero, quasi-BICs appear from the $E_2^{(1)}$ and $E_2^{(2)}$ modes.

Lattice deformations to control mode spacing and Q -factors

To show a single lattice deformation capable of controlling mode spacing and Q -factors we performed full-wave eigenvalue simulations when the symmetry breaking operation was shifting alternating rows of resonators. A schematic of this symmetry breaking operation is presented in Fig. S8a. Shifting alternating rows of resonators causes all symmetries to be broken resulting in a lattice belonging to the $p1$ space group. This symmetry reduction, results in the symmetry-protected degenerate BICs in becoming non-degenerate quasi-BICs. This behavior is confirmed by the full-wave eigenvalue simulations. As the resonator deformation is increased from 0 nm the degeneracy is lifted (Fig. S8b)

and the Q -factors (Fig. S8c) drop from infinity.

Figure R3: **Shifting alternating rows of resonators to control splitting and Q -factors** (a) Schematic illustrating shifting alternating rows of cylindrical resonators. This deformation breaks all symmetries reducing the the symmetry from a lattice belonging to the $p6mm$ space group to one belonging to the $p1$ space group causing the symmetry-protected degenerate BICs to become non-degenerate quasi-BICs. This effect can be observed in the wavelengths (b) and Q -factors (c). As expected, when the displacement is nonzero the degeneracy is lifted and the Q -factors become finite.

Independent control of splitting and Q -factor

To demonstrate full control of Q -factor and splitting, we designed and fabricated a series of silicon metasurfaces on fused silica substrates where two classes of deformations were applied to break the \$C_2\$ and \$C_3\$ symmetries ~~-.The fabricated-~~ of a lattice that would otherwise possess \$C_{6v}\$ symmetry. The silicon metasurfaces had a period - 1000 nm, thickness - 200 nm, resonator diameter - 800 nm, and bar width - 100 nm. ~~A false-color scanning electron micrograph of a representative metasurface is presented in Fig. 2a.~~ While there are many combinations of resonator and lattice deformations capable of breaking \$C_2\$ and \$C_3\$ symmetries, a resonator and lattice deformation were chosen to break \$C_2\$ and \$C_3\$ respectively since the combination is more compatible with the capabilities of electron-beam lithography. An extended discussion on design considerations and an experimental demonstration using only resonator deformations to break \$C_2\$ and \$C_3\$ symmetries are presented in the supplementary information.

To break C_2 symmetry, the circular resonators were smoothly deformed into triangular resonators (geometric definition in Methods) as depicted in Fig. 2b-a (this operation preserves C_3 and reflection symmetries, as well as the filling fraction of the unit cell). In contrast, the lattice was contracted along one axis to break C_3 symmetry (an operation that preserves C_2 and reflection symmetries). From these deformations, we define the C_2 and C_3 symmetry breaking parameters as S and δ respectively. A false-color scanning electron micrograph of a representative metasurface is presented in Fig. 2b.

2.2. Comment #2

RC: *Starting from Section “Independent control of splitting and Q-factor” theoretical and experimental results are described in confusion way. A small part of text is to experiment and the very next part is theory without any warning and right at once it goes back to the experiment and so on. It misleads a reader and has to be rewritten in a more structuralized way*

AR: We thank the reviewer for identifying how the structure unnecessarily reduces clarity. To improve clarity we have updated the manuscript and changed the structure of the section "Independent control of splitting and Q-factor" to move from design to simulations and ending with experimental results. Because of the long length of the section we have not reproduced the updated section below.

2.3. Comment #3

RC: *Oblique incidence is known to cause reduction of the symmetry. Is a one degree convergence is negligible. The authors have to carry out proper simulations sustaining this assumption.*

AR: We thank the reviewer for identifying this potential issue. We agree that oblique incidence can cause symmetry reduction, and therefore we have performed full-wave electromagnetic simulations to determine the effect of oblique incidence. We have added a section to the supplementary information to discuss these simulation and compare predicted mode splittings with experimentally measured mode splittings. From this analysis, we find that nonzero angle of incidence is not the largest contributor to symmetry breaking when the degeneracy is purposely lifted through design. The new section in the supplementary information has been reproduced below for the reviewer’s convenience.

Effect of nonzero angle of incidence

In the experimental measurements, the angle of incidence was maintained less than 1° . Nonzero angle of incidence results in symmetry breaking. To determine the effect of angle of incidence on symmetry reduction, we performed full-wave electromagnetic simulations of reflectance (Fig. S9a) when the resonator symmetry breaking (S) was 0.4 and lattice deformation (δ) of 0 nm for angles ranging from 0° to 3° . From the calculated reflectance spectra, as the angle of incidence is increased the induced symmetry breaking results in the degeneracy being lifted and a reduction of the Q -factors. The calculated mode splitting (Fig. S9b) was recovered from the simulated reflectance curves by fitting the spectra with two Fano resonances. When the angle of incidence reaches 1° the mode splitting increases to 0.3 nm. Increasing the angle of incidence further to 3° results in a splitting of 2.7 nm. These calculated mode splittings are below the experimentally measured mode splittings for the half-circle, half-square resonator deformation case which observed splittings of 4 nm and 10 nm demonstrating that nonzero angle of incidence is not the largest contributor to symmetry breaking. For additional confirmation, we fit the experimentally measured reflectance measurements (Fig. S9c) of degenerate quasi-BICs when the resonator symmetry breaking (S) was 0.4 and lattice deformation (δ) of 0 nm. Using a two Fano resonance fit resulted in a mode splitting of 1.5 nm representing an upper bound on the effect of angle of incidence since symmetry breaking can be caused by both nonzero angle of incidence and fabrication imperfections. This observed mode splitting would correspond to an angle of incidence of 2.2° for the case when nonzero angle of incidence was the only source of symmetry breaking.

Figure R4: **Effect of nonzero angle of incidence** (a) Calculated reflectance spectra when the angle of incidence is nonzero. (b) To calculate the effect of angle of incidence on mode splitting, the calculated reflectance spectra were fit with two Fano resonances. As the angle of incidence was increased from 0° the degeneracy was lifted. (c) Representative experimentally measured reflectance spectra for the case when the modes are degenerate quasi-BICs ($S=0.4$, $\delta=0$ nm). Fitting the reflectance spectra with two Fano resonances results in a mode splitting of 1.5 nm providing a metric for the amount of symmetry breaking from nonzero angle of incidence and fabrication imperfections.

2.4. Comment #4

RC: *The manuscript contains vast of the group-symmetry terms, which are in excessive way and sometimes misleading. I recommend the authors to give their manuscript to an expert in the group theory analysis applied to condensed matters to improve the terminology.*

AR: The group theory notation we use in our manuscript is chosen to be identical to that of “Group theory and its applications in physics,” by Inui, Tanabe, and Onodera, Ref. [55] in the manuscript, a standard textbook for group theory across all of physics, and in particular in condensed matter physics. Since it is necessary to keep track of space group symmetry in addition to point group symmetry to determine the symmetry of the perturbed field, we have adopted the abbreviated Hermann–Mauguin notation used by the International Union of Crystallography (IUCr), which is consistent with other published works in this field. However, we agree that this standard notation can be slightly frustrating, and so we have endeavored to clarify the language surrounding this notation to improve the manuscript’s readability.

This has resulted in numerous changes throughout the manuscript’s abstract and main text.

2.5. Comment #5

RC: *Several times the authors claim on benefits over the problem of fabrication imperfections, however they refer to no studies. For example, recent paper in Nanophotonics 10, 4313 (2021) examines the robustness of the BIC of different type against the typical fabrication imperfectness, that is positional disorder. I recommend the authors to consider that paper and references therein.*

AR: We thank the reviewer for pointing out this oversight. We have added appropriate references when discussing the susceptibility of BICs to disorder.

Although some recent studies have attempted to overcome this shortcoming through fine-tuning an accidental degeneracy between two otherwise spectrally isolated BICs^{36,45–49}, ~~this approach results in systems that yielding nearly degenerate states where the frequency splitting, center frequency, and relative polarization states in such systems~~ are extremely sensitive to material variations and fabrication imperfections, both of which yield unpredictable changes ~~in the frequency splitting between, and the Q-factors of, to the properties of~~ the two states^{50–52}

2.6. Comment #6

RC: *Overall, I like the idea but the present form of the manuscript makes me unhappy. My recommendation is a major revision mainly to improve the presentation of the results and to make it clearer.*

AR: To address the reviewer's comments we have made major revisions to the manuscript. Broadly speaking, these changes included adding discussions and results to demonstrate C_2 -breaking and C_3 -breaking controlling the Q-factors and mode splitting, respectively, which clarifies the design paradigm. Furthermore, we have reorganized the theoretical and experimental results for increased clarity.

3. Reviewer #3

RC: *In the manuscript, Authors discuss the origin of bound states in the continuum, high-Q resonances of periodic photonic structures, in terms of group theory. In particular, they present an approach to create two degenerate symmetry-protected bound states in the continuum and control their quality factor. It is a beautiful example of rigorous theoretical analysis which results can be implemented in practical applications.*

AR: We want to thank the reviewer for their time and constructive feedback. The feedback enabled us to significantly strengthen the manuscript.

3.1. Comment #1

RC: *Line 145 seems to have a misprint.*

AR: Thank you for pointing out this awkward wording. We have modified the section in the course of addressing Reviewer #3, Comment #2. Please see Comment #2 for the revised text.

3.2. Comment #2

RC: *In Fig.3(a) diagram shows different regimes of symmetry breaking. How was the quality factor obtained? Highly-likely, from simulation, but why did the Authors take average values of quality factor?*

AR: We thank the reviewer for identifying this ambiguity. The reviewer is correct that the Q-factors were obtained from simulations. In the non-degenerate BIC and degenerate quasi-BIC phases the Q-factors for both modes are required by symmetry to be identical. In comparison, when in the non-degenerate quasi-BIC phase no such requirement exists and the Q-factors of the two modes can differ. To keep improve the readability of Fig. 3a we only plotted the mean Q factor which creates unnecessary ambiguity. We have modified Fig. 3a so that the mean, minimum, and maximum Q-factors are plotted. We feel the new plot maintains readability while significantly improving clarity. We have modified the manuscript text to more clearly describe these considerations to the reader. The updated text and figure are reproduced below for the reviewer's convenience.

When both lattice and resonator deformations are present ($\delta > 0$ nm and $S > 0$), the triangular lattice is reduced to a lattice belonging to the cm space group. Under this symmetry reduction, the degeneracy is no longer guaranteed by symmetry and the modes can couple to the radiative continuum in the environment. This means that the modes in this phase can be characterized as non-degenerate quasi-BICs with Q -factors and splitting controlled by lattice and resonator deformations respectively. (Specifically, the E_2 irreducible representation is reduced into the 1-dimensional A and B irreducible representations.) Since both C_2 and C_3 rotational symmetries are broken, the two quasi-BICs are no longer required to have the same Q -factors. Because of this, Fig. 3a presents the mean, maximum, and minimum Q -factors for the two $E_2^{(1)}$ modes of a silicon metasurfaces. As the lattice deformation is increased the difference between the Q -factors increases.

Figure R5: **Symmetry phases for engineering splitting and Q -factors.** (a) Symmetry phase diagram for deforming resonators into triangles and contracting the lattice. The deformations determine the splitting and Q -factors for the $E_2^{(1)}$ modes of a silicon metasurface. When the resonator is deformed into a triangle, only C_2 symmetry is broken resulting in two degenerate quasi-BICs. The Q -factors can be controlled by the resonator symmetry breaking parameter (S). Correspondingly, when the lattice is contracted in one direction two non-degenerate BICs are formed since only C_3 symmetry is broken. With these two symmetry breaking parameters it is possible to design a lattice with cm C_s symmetry capable of having non-degenerate quasi-BICs with arbitrary splitting and Q -factors. (b) Field profile for the four symmetry phases.

3.3. Comment #3

RC: *Could the Authors comment on how the orientation of the axis along which the lattice is squeezed, is chosen? Is it restricted by basis vectors of the primitive unit cell? Will there be any quantitative differences in Q s and frequencies splitting compared to the discussed case?*

AR: We thank the reviewer for bringing up this question. While the lattice can be squeezed along any axis that reduces the lattice to a centered rectangular Bravais lattice, the resulting Q -factors and mode splittings can differ significantly. To address this issue, we have added a section the supplementary information discussing the effect of lattice contraction direction on mode splitting and Q -factors. The section is reproduced below for the reviewer's convenience.

Effect of lattice contraction direction

To understand the role of lattice contraction direction on mode splitting and Q -factors we performed full-wave eigenvalue simulations for three lattice contraction directions: vertical, 63.4° diagonal, and horizontal. A schematic of the three contraction directions with respect to the triangular lattice is presented in Fig. 10a. For all cases, the C_2 symmetry breaking parameter (S) was 0.6. The calculated Q -factors and mode splittings (Fig. 10b) show that the choice of lattice contraction direction can have three quantitative effects. First, for a given mode splitting the mean Q -factors can depend strongly on contraction direction. For example, when the splitting was near 25 nm, the mean Q -factors (\bar{Q}) for the vertical and horizontal cases are 2700 and 2730 respectively. In comparison, for the diagonal contraction case the calculated Q -factors were significantly higher with a \bar{Q} of 3380. The second effect is the difference between the maximum (Q_{max}) and minimum (Q_{min}) Q -factors ($\Delta Q = |Q_{max} - Q_{min}|$). For the diagonal contraction case, the calculated ΔQ when the mode splitting is close to 25 nm is 1110. This difference reduces to 980 for the horizontal contraction case. The smallest ΔQ is obtained for the vertical contraction with a difference of 920. The final quantitative effect of lattice contraction direction is the degree of symmetry breaking for a given amount of lattice contraction. For the diagonal contraction case, the unit cell area has been reduced the most, but only achieves a mode splitting of around 25 nm. While the unit cells are larger for the vertical and horizontal cases, they achieve mode splittings of 45 nm and 41 nm respectively. The reduced splitting in the diagonal case can be attributed to the fact that the diagonal lattice contraction is closer to preserving the triangular lattice (albeit with a scaled periodicity). If the lattice contraction direction only scaled the periodicity of the triangular lattice, C_3 symmetry would be preserved and the modes would remain degenerate. Combined, these three quantitative effects from lattice contraction direction demonstrate how the choice of symmetry breaking method can significantly affect device performance and provide insight into potential routes to optimize symmetry breaking operations.

Figure R6: **Effect of lattice contraction direction on Q -factor** (a) Schematic denoting three different directions for contracting the triangular lattice. (b) Calculated Q -factors from full-wave eigenvalue simulations for vertical, horizontal, and diagonal lattice contractions. For equivalent mode splitting values the mean Q -factor (\bar{Q}) and difference between Q_{max} and Q_{min} can differ significantly.

References

- [1] Teturo Inui, Yukito Tanabe, and Yositaka Onodera. *Group Theory and Its Applications in Physics*. Springer Science & Business Media, December 2012.
- [2] Tomás Santiago-Cruz, Sylvain D. Gennaro, Oleg Mitrofanov, Sadvikas Addamane, John Reno, Igal Brener, and Maria V. Chekhova. Resonant metasurfaces for generating complex quantum states. *Science*, 377(6609):991–995, August 2022.

REVIEWERS' COMMENTS

Reviewer #1 (Remarks to the Author):

Authors scrupulously answered all questions and comments of my review. The revised paper definitely can be published in Nature Communications. In conclusion I'd like pay attention of authors to recently published paper in PRB 106, 085404 (2022) by A. Sadreev et al in which similar symmetry aspects for degenerated BICs were issued in acoustic resonators of C_{4v} and C_{3v} symmetries.

Reviewer #2 (Remarks to the Author):

I read the authors' response. I find the manuscript improved in accordance with Reviewers' recommendations and now it is suitable for publication in NCOMM

Reviewer #3 (Remarks to the Author):

Dear authors,

thank you for detailed and clear comments, significant improvement of the text. I do not have any other questions. I am glad that you have found my questions useful.

Authors' Response to Reviews of

Realizing symmetry-guaranteed pairs of bound states in the continuum in metasurfaces

Chloe F. Doiron, Igal Brener, and Alexander Cerjan
Nature Communications,

RC: Reviewers' Comment, **AR: Authors' Response**, Manuscript Text

1. Reviewer #1

RC: *Authors scrupulously answered all questions and comments of my review. The revised paper definitely can be published in Nature Communications. In conclusion I'd like pay attention of authors to recently published paper in PRB 106, 085404 (2022) by A. Sadreev et al in which similar symmetry aspects for degenerated BICs were issued in acoustic resonators of C_{4v} and C_{3v} symmetries.*

AR: We want to thank the reviewer one last time for their time and constructive feedback. We agree that the paper is noteworthy, and have added the reference to the manuscript to highlight the potential for BICs in acoustic systems.

2. Reviewer #2

RC: *I read the authors' response. I find the manuscript improved in accordance with Reviewers' recommendations and now it is suitable for publication in NCOMM*

AR: We want to thank the reviewer one last time for their time and constructive feedback.

3. Reviewer #3

RC: *Dear authors, thank you for detailed and clear comments, significant improvement of the text. I do not have any other questions. I am glad that you have found my questions useful.*

AR: We want to thank the reviewer one last time for their time and constructive feedback.